# EEG Features in Autism Spectrum Disorder: A Retrospective Analysis in a Cohort of Preschool Children

**DOI:** 10.3390/brainsci13020345

**Published:** 2023-02-17

**Authors:** Marta Elena Santarone, Stefania Zambrano, Nicoletta Zanotta, Elisa Mani, Sara Minghetti, Marco Pozzi, Laura Villa, Massimo Molteni, Claudio Zucca

**Affiliations:** 1Psychopathology Department, IRCCS E. Medea, 23842 Bosisio Parini, Italy; 2Clinical Neurophysiology Unit, IRCCS E. Medea, 23842 Bosisio Parini, Italy; 3Scientific Institute IRCCS E. Medea, 23842 Bosisio Parini, Italy

**Keywords:** autism, EEG, paroxysmal abnormalities, epileptiform abnormalities, sleep

## Abstract

Autism Spectrum Disorder (ASD) is a complex neurodevelopmental disorder that can be associated with intellectual disability (ID) and epilepsy (E). The etiology and the pathogenesis of this disorder is in most cases still to be clarified. Several studies have underlined that the EEG recordings in children with these clinical pictures are abnormal, however the precise frequency of these abnormalities and their relationship with the pathogenic mechanisms and in particular with epileptic seizures are still unknown. We retrospectively reviewed 292 routine polysomnographic EEG tracings of preschool children (age < 6 years) who had received a first multidisciplinary diagnosis of ASD according to DSM-5 clinical criteria. Children (mean age: 34.6 months) were diagnosed at IRCCS E. Medea (Bosisio Parini, Italy). We evaluated: the background activity during wakefulness and sleep, the presence and the characteristics (focal or diffuse) of the slow-waves abnormalities and the interictal epileptiform discharges. In 78.0% of cases the EEG recordings were found to be abnormal, particularly during sleep. Paroxysmal slowing and epileptiform abnormalities were found in the 28.4% of the subjects, confirming the high percentage of abnormal polysomnographic EEG recordings in children with ASD. These alterations seem to be more correlated with the characteristics of the underlying pathology than with intellectual disability and epilepsy. In particular, we underline the possible significance of the prevalence of EEG abnormalities during sleep. Moreover, we analyzed the possibility that EEG data reduces the ASD clinical heterogeneity and suggests the exams to be carried out to clarify the etiology of the disorder.

## 1. Introduction

Autism Spectrum Disorder (ASD) is a complex neurodevelopmental disorder characterized by persistent deficits in social communication and interaction and by restricted, repetitive patterns of behavior, interests and/or activities. It becomes clinically evident during the early developmental period and leads to different degrees of impairment in personal, social and occupational functioning depending on the spectrum and severity of symptoms [1].

In subjects with ASD, cognitive skills range widely; some individuals present with severe intellectual disability, others show high-level cognitive functioning. Neurological manifestations such as epilepsy, motor impairment and sleep disorders are common, and also associated with the more severe phenotypes of ASD. Thus, the early diagnosis and treatment of these comorbidities could improve the behavioral and cognitive difficulties [2].

Based on the latest Centers for Disease Control and Prevention’s report (2018), in the US, 1 in every 44 children aged 8 years is affected by ASD, with an average ratio 4:1 for male: female and with an increasing trend of prevalence in recent years [3]. In Italy 1 in 87 children aged 7–9 years is affected by this disorder [4]. 

Despite symptoms could appear in the first year of life, diagnosis of ASD is usually established in children between 2 and 4 years, according to DSM-5 clinical criteria, with the support of behavioral tests and parental interviews [1,5].

A number of environmental and biological factors may potentially contribute to the etiopathogenesis of the disorder. Most of them, such as congenital infections or toxicant exposures, act during fetal and perinatal periods, in the developmental phase of the central nervous system. A strong genetic component in ASD has also been confirmed [6].

In the study by Donovan et al., (2017), several neuroanatomical abnormalities have been identified in brain structures and neuronal circuits such as cerebellum, limbic system, corpus callosum and frontal cortex of subjects affected by the disorder [7], supporting an abnormal neural connectivity behind ASD, with a consequent atypical brain development [5]. To date, no certain biological marker of autism has been identified; nevertheless, some authors propose the EEG findings as possible predictive elements of the developmental and the adaptive functioning in ASD [8,9,10].

An abnormal EEG characterizes 8–80% of non-epileptic ASD children and a mean of 39.1% of subjects with autism of all ages with or without epilepsy [11,12]. About 45.5% of individuals show EEG paroxysmal abnormalities, mainly in childhood and in females, while in the general population they occur in about 2–8.7% of subjects with a decreasing trend during puberty [13]; 4–61% of these individuals specifically present interictal epileptiform discharges (IEDs), mainly focal, in particular over frontal and temporal areas [14]. About 5–40% of people with ASD suffer from epilepsy, with a mean of 24.9% [13,15]. In Hrdlicka et al., autistic regression is reported to occur more frequently in subjects with epilepsy (*p* < 0.01), as well as an abnormal neurodevelopment during the first year of life to be related with epileptiform EEG abnormalities (*p* < 0.05) and epilepsy to be significantly associated with intellectual disability (*p* < 0.01) [16]. In Nicotera et al. (2019), EEG abnormalities are stated to correlate with several phenotypic features of ASD, even in the absence of epilepsy (*p* < 0.05 or *p* < 0.01): severe forms of autism, behavioral problems, motor stereotypies, intellectual disability and language impairment [12]. 

Within this context, the primary aim of our study is to classify and quantify the EEG abnormalities recorded in a cohort of preschool children with ASD undergone to polysomnographic EEG (PSG-EEG) study at the IRCCS E. Medea, Bosisio Parini.

Secondary aim is to estimate the prevalence of children with Paroxysmal Slowing and Interictal Epileptiform Discharges (PS-IEDs) on EEG recordings, both during wakefulness and sleep, to highlight any difference in their occurrence and to determine the prevalence of epilepsy within the same group.

Tertiary aim is to explore a possible relationship between the EEG findings and the clinical phenotype of individuals with ASD, in term of intellectual disability, considering the potential usefulness of EEG data in the clinical classification, diagnostic and therapeutic management of these subjects.

## 2. Materials and Methods

The data presented in this study were collected at the IRCCS E. Medea (Bosisio Parini, Italy) a Scientific Institute funded by Italian Ministry of Health, dedicated to diagnosis, treatment and biomedical research of neurodevelopmental, neurological and neurodegenerative diseases of pediatric age. 

### 2.1. Clinical and Instrumental Investigations in ASD Subjects

Clinical diagnosis of ASD was established in the children according to DSM-5 criteria by a multidisciplinary team [1], constituted by specialists trained in evaluation of neurodevelopmental disorders. The clinical diagnosis was confirmed by certified psychologists/psychiatrists through the administration of Autism Diagnostic Observation Schedule, Second Edition (ADOS-2) Italian Version [17] in all subjects and the Autism Diagnostic Interview Revised (ADI-R) Italian Version [18] when needed. To complete the diagnostic evaluation, children underwent a cognitive/developmental assessment through standardized tests (Griffiths Mental Development Scales, Revised—GMDS-R and –ER, Italian Editions—or Wechsler Preschool and Primary Scale of Intelligence, III Italian edition—WPPSI-III) [5,19]. According to the Italian Guidelines for Autism [20] and to the Guidelines of American Academy of Pediatrics [21] in all the subjects further clinical and instrumental investigations were performed, including PSG-EEG, genetic clinical evaluation, genetic tests (karyotype, FMR1 analysis and/or Array-CGH). Brain MRI was performed in selected cases, i.e., when suggested by clinical history, neurological evaluation and clinical data (perinatal distress, history of regression, neurological signs).

### 2.2. Polysomnographic Recordings and Analysis

At the IRCCS E. Medea in all children of early/pre-school age admitted to the Psychopathological Department, in which ASD has been diagnosed, a PSG-EEG is recorded during daytime spontaneous sleep, with average recording time of 40–70 min. This protocol is based on the Clinical Practice Guidelines by SINPIA and APA [21]. These guidelines establish that children with ASD diagnosis should undergo EEG recording when: the child has an important speech delay; a behavioral or language regression is reported; when seizures occur or paroxysmal events are suspected (such as episodes of unresponsiveness). This protocol is also sustained by meta-analysis of Worldwide Clinical Practice [22]. Based on these recommendations, a routine PSG-EEG recording is performed in all the children of early/pre-school age at diagnosis.

Parents are asked for consent to use of the data, collected through EEG recordings and other instrumental diagnostic tests, for research purposes in anonymous and aggregated form.

In the light of the abovementioned recommendations and guidelines, we present in this paper a retrospective analysis of the PSG-EEG recordings collected over 30 months in order to describe the amount and characteristics of the abnormal data and explore the usefulness of this procedure in the diagnosis of ASD at an early age.

All the PSG-EEG were digitally acquired (by MICROMED BQ32SYNCMASTER) at the Clinical Neurophysiology Unit of IRCCS E. Medea, with 20 scalp electrodes (placed according to the international standard 10–20 system) and at least 2 surface electromyographic (EMG) electrodes (mainly placed over deltoids muscles of both sides).

The recordings were analyzed by 3 expert clinical neurophysiologists, unaware of subjects’ clinical and diagnostic details.

In all the recordings, features of the background activity during wakefulness and sleep were analyzed. Moreover, the presence and the location (focal or diffuse) of abnormal slow waves and PS-IEDs were assessed and classified according to the definitions of the International Glossary of clinical electroencephalography [23].

### 2.3. Inclusion and Exclusion Criteria

We included in the study all children, younger than 6 years, who performed EEG-PSG recording in the context of diagnostic evaluation for ASD and in whom we obtained a sleep recording. 

All children in whom ASD was associated with a defined neurological and/or metabolic or genetic disease with a documented etiology were excluded.

### 2.4. Statistical Analysis

We carried out exploratory statistical analyses only on categorical data, with contingency tables and one-tailed chi-square tests to evaluate the possible associations between PS-IEDs and neurodevelopmental delay and between neurodevelopmental delay and abnormal background activity at the EEG. We also calculated the prevalence of subjects with PS-IEDs only during sleep or even during wakefulness. We performed all data analysis with GraphPad Prism 8 (for Windows 64-bit). A probability (*p*)-value less than 0.05 (≤0.05) was considered statistically significant.

## 3. Results

In 30 months we analyzed PSG-EEG recordings in 298 children at the Clinical Neurophysiology Unit of our Institute.

We excluded 6 subjects from the study because they met the exclusion criteria: 5 children did not sleep during the recording (only drowsiness was recorded in 1 child and only wakefulness in the other 4). One child was diagnosed during the clinical workup with a type 1 Neurofibromatosis. Thus, 292 PSG-EEG recorded in 292 children with ASD were examined (Table 1). Mean age at recording time was 34.6 months (age range: 18.8–56.7 months). Sex distribution was the following: 84.9% of children were males and 15.1% females, with a ratio of 6:1. In our cohort, only a minor percentage of girls had a normal development. 

No child showed focal neurological signs, but 190 children (69.5%) presented with developmental delay. Only 1 individual (0.35%) received a diagnosis of epilepsy (focal epilepsy of unknown etiology). Twenty-four subjects underwent MRI and abnormal findings were found in 15 cases (examples in Figure 1). Among these cases, 7 showed posterior fossa abnormalities (1 vermian hypoplasia, 2 Arnold-Chiari type 1 malformation, 1 mild fourth ventricle enlargement, 3 mild cisterna magna enlargement), 3 had a ventricular asymmetry, 2 a slight enlargement of the perivascular spaces, 1 focal heterotopia, 1 mild white matter signal abnormalities, 1 mild white matter signal and macrocephaly.

During wakefulness, slow or irregular background activity was recorded in 58.0% of the subjects, asymmetry in 21.0% and abnormal fast activity in 23.0% of the cases. Abnormal slow waves were localized over the occipital regions in 40.0% of subjects, over the central regions in 30.0% of cases and over the frontal ones in the remaining 30.0% of cases.

In our cohort, during non-REM 1 and non-REM 2 sleep stages (example in Figure 2), slow or irregular background activity was found in 78.0% of cases; asymmetry in 9.0%, asynchrony in 10.0% and abnormal fast activity in 12.0% of the subjects. Focal slow abnormalities were localized over the central regions in 70.0% of children.

In our sample, 83 out of 292 of the EEG recordings (28.4%) showed PS-IEDs.

In 79 cases (95.2%) PS-IEDs were present only during sleep and in 4 cases (4.8%) also during wakefulness, specifically with a frequency 19.7 times higher during sleep. As far as the localization of PS-IEDs is concerned, 37.7% of the PS-IEDs were focal and 62.7% bilateral, diffuse. Focal PS-IEDs were in 48.4% of recordings over central areas, 32.3% over temporal areas and 19.3% over the frontal ones.

During wakefulness, most of children (96.0%) did not show EEG alterations, while a minor percentage (4.00%) showed PS-IEDs or abnormal slow waves. On the other hand, in 76.0% of the subjects, slow or paroxysmal abnormalities were found during sleep. Examples are showed in Figure 2, Figure 3, Figure 4 and Figure 5.

We explored the relationship between developmental level impairment, abnormal features of background activity in sleep and the presence of PS-IEDs.

As far as PS-IEDs are concerned, a developmental delay was found in 68.7% of individuals with PS-IEDs and in 64.1% of subjects without PS-IEDs with no significant association between the two factors: *p* = 0.2787.

However, abnormal background activity during sleep was associated to developmental delay in 73.5% of the individuals, while normal background activity was related to developmental delay in 54.6% of the cases, with an extremely significant association between the two factors: *p* = 0.0004.

## 4. Discussion

There is a growing interest towards EEG features of ASD subjects, with particular focus on background abnormalities and IEDs, despite the lack of concluding evidence on the diagnostic, prognostic and therapeutic role of EEG data in ASD. Nevertheless, EEG features could be particularly useful in reducing heterogeneity in this spectrum of neuropsychiatric developmental disorders. 

PS-IEDs are frequent among ASD people, while are rare (1–4%) in healthy subjects [24]. Several studies have reported a rate of approximately 30% of PS-IEDs in ASD individuals with no clinical evidence of epileptic seizures [16,25,26]. These rates vary from 5% to 46%, due to sample and methodological variability in collecting and interpreting EEGs [27]. These abnormalities are considered a nonspecific sign of cortical dysfunction, even in absence of clinical epilepsy and some authors propose that they may concur to determine the autism phenotype [28]. 

Results of our study confirm the high amount of EEG abnormalities in ASD children, with 28.4% of subjects showing PS-IEDs during sleep, while only a minor percentage showing PS-IEDs during wakefulness. These data are aligned with others studies reporting a prevalence of PS-IEDs of about 30% in a similar sample, i.e., children with ASD without epilepsy (mean age about 3 years) [26,29]. 

As already highlighted in some papers [30,31], the non-REM sleep stages are those in which the subjects’ recordings were more informative. In addition to that, our study highlights how PS-IEDs were recorded in the same percentage of individuals with and without developmental delay, suggesting that ASD itself may be associated with these EEG features. 

In literature, abnormal background activity is statistically associated to developmental delay [32,33]. In our study, background activity abnormalities were found in 58% of our cases during wakefulness and in 78% during sleep. These data suggest that PS-IEDs rather than background abnormalities may be considered as a neurophysiological marker of ASD, supporting the hypothesis that ASD may be due to aberrant neural circuitry of specific cortical and sub-cortical structures [7].

Particularly, the relevant finding of PS-IEDs mainly during sleep recordings, may suggests that these abnormal circuitries could be linked to the physiological, cyclical pattern of the neuronal firing during non-REM sleep [27], confirming the importance to analyze the EEG data in subjects with ASD necessarily during sleep.

With our data we don’t want to suggest that the visual analysis of the EEG constitutes directly a diagnostic tool. Indeed, the diagnosis of ASD is a clinical one, based on the analysis of the developmental history and the observation of the child’s interaction with their parents and unfamiliar adults during a combination of structured and unstructured assessments [34].

Nevertheless, the correlation of sleep EEG data with a clinical phenotype, such as the presence of developmental delay, can help in facilitating the definition of neurological endophenotypes with a reduction of the heterogeneity in ASD study samples and may also be a clinical marker. The detection of focal EEG abnormalities recommends MRI investigations in individuals with ASD. Surely, a bias of our study is the low percentage of MRI performed, due to the fact that the above recommendation has not yet entered the diagnostic routine of the clinical practice. 

Other findings, such as excess of fast activity, asynchronous PS-IEDs over the central regions of the brain, detection of non-epileptic myoclonus, should prompt clinicians to perform tests aimed at identifying a possible genetic cause [35].

In our sample, only one subject was diagnosed with an epileptic syndrome. This rate (0.35%) is very low compared to data reported in literature, although many children of our sample showed intellectual disability (66%), which is considered as a risk factor for developing epilepsy in ASD. Indeed, two peaks of epilepsy onset are described in ASD so far [36]: the first peak during early childhood (with lower frequency), the second and most significant one during adolescence-early adulthood. Our cohort is characterized by an early age at diagnosis (children with ASD younger than 6 years; mean age 34.6 months) justifying both the very low percentage of epilepsy and the unexpected sex ratio. In fact, in the population of children affected by epilepsy followed at our Epilepsy Center, those with ASD comorbidity are mostly over 6 years of age. Another reason is that our subjects are addressed to the Psychopathology Department with ASD as referral suspect. Therefore, there could be a selection bias whereby children of such early age already affected by epilepsy do not refer to the Psychopathology Department of our Institute for the autistic syndrome diagnosis.

As far as the unexplained M:F ratio is concerned, it may be related to the fact that females have been found to be diagnosed with ASD at significantly later ages and to experience greater delays in the time from an initial evaluation to receiving a clinical ASD diagnosis [37,38,39]. Qualitative self-report data have also supported the theory that females may be under-diagnosed, as many females who were diagnosed with ASD late (i.e., in adolescence or adulthood) report that they received a series of inaccurate diagnoses prior to their ASD diagnosis [40,41,42]. Moreover, it is also well-established that the sex imbalance in prevalence varies with cognitive ability, with a smaller male to female ratio of approximately 2:1 among individuals with co-occurring ID and a much larger ratio of as much as 6:1 among those with average to above average IQ [43,44,45,46]. This pattern may indicate that as the autism spectrum has expanded to include more individuals without co-occurring ID, females in this group have not been adequately identified. As we reported in our results, in our cohort, most of the females have ID (85%), in contrast to a lower percentage of males (60%). Given this mounting evidence that ASD is not adequately identified in females without co-occurring ID, many have theorized that there may be sex differences in the manifestation of autistic traits in this group, which could in turn contribute to diagnostic disparities [47,48]. Autistic females without ID tend to show more developmentally appropriate vocabulary and core language skills than their male counterparts [49,50,51] though not always [52]. Given that language delays are the most commonly reported first concern among parents of children with ASD [53], this difference may have important implications for diagnostic timing and accuracy. Autistic females without ID are also more likely to have intact play and imitation skills, which are often considered core impairments in ASD [44,54,55]. Additionally, in contrast to the social isolation classically described among autistic boys, girls are more likely to be described as “clingy” or overly concerned with being liked by peers [44,50]. As described above, the camouflaging effect may mean that there are some autistic females who are not identified by current measures and thus are excluded from studies as ours.

The use of anti-seizures medications in individuals with PS-IEDs in the absence of clinical seizures is controversial [56]. As we said, if a common cause for both disorders is hypothesized, the treatment seems not advisable. On the other hand, if PS-IEDs are believed to cause developmental impairment, a drug-based treatment may be justified [56], particularly when PS-IEDs occupy a significant proportion of non-REM sleep and an arrest of the neuro-cognitive development is detected simultaneously.

Finally, the prognostic significance of PS-IEDs in young subjects with ASD on the possibility of later developing epileptic seizures can be only clarified with prospective studies with quite long follow-up.

Another possible use of EEG data in subjects with ASD is the detailed study of the prognostic questions regarding the outcome of autism.

ASD has a heterogeneous developmental time course [57]. Indeed, subgroups of individuals with autism and improving or worsening symptoms over time can be identified [57]. Furthermore, co-occurring disorders may contribute to autism severity in the context of this heterogeneity. The prevalence of this co-occurring condition can vary considerably [57] and PSG-EEG data may represent a tool to explore this prevalence.

Several studies have explored the relationship between EEG abnormalities and adaptive behavior, executive functioning, severity of ASD core symptoms and other psychiatric symptoms, finding that [58] children with epileptiform abnormalities performed worse on executive functioning assessments and exhibited higher scores in inhibition self-control compared with children without epileptiform abnormalities. At the same time, they presented lower adaptive functioning in the setting of ASD.

Therefore, EEG abnormalities, especially in early age, can have a negative impact on brain development, compromising both cognition and behavior [58].

For this reason, several rehabilitation projects have been launched in our Institute on very young children who receive a diagnosis of ASD [57].

Our data represent a starting point for setting up further prospective studies aiming correlating over time the outcome of these therapeutic procedures with PSG-EEG findings at the time of diagnosis.

These studies may outline the profiles of children most at risk of having a worse prognosis and a lower response to rehabilitation interventions.

Our study presents with some limitations. The first one is due to the initial selection of the subjects included. In order to better homogenize our cohort, we choose to include in our study only preschool children. Focusing on this early age, may have missed data on the development of epilepsy in older age.

Moreover, clearly syndromic individuals and epileptic subjects are less likely to come to our attention as a first step.

Based on these limitations, we chose to conduct only an exploratory analysis, we did not calculate sample size and we did not deal with confounding factors and other clinical variables, which we will examine in future perspective studies. However, we calculated study power post-hoc on the main outcome analysis, i.e., the contingency table crossing epileptiform abnormalities X delayed development, obtaining 10.9% power. Therefore, we can conclude that this study is not adequately powered to claim that there is no difference in the occurrence of epileptiform abnormalities between children with delayed and normal development. We calculated study power post-hoc also on the secondary outcome analysis, i.e., the contingency table crossing background activity X delayed development, obtaining 92.2% power.

Our intention is to proceed with a detailed characterization of behavior, language, type of delay, adaptive skills, functionality and to perform a multivariable correlation, in a larger cohort, between the clinical and the neurophysiological data. Additionally, we will thoroughly characterize the dysmorphological, genetic and abnormal neuroradiological findings and their correlation with EEG data.

## 5. Conclusions

Our study suggests that there is a high rate of EEG abnormalities in preschool children diagnosed with ASD. These alterations concern both features of the background activity and, most importantly, the presence of PS-IEDs. These EEG abnormalities seem to be much more noticeable in sleep than in wakefulness.

If confirmed, these preliminary results could be particularly useful to better define the heterogeneity of ASD pictures and to direct further investigations, particularly genetic and neuroradiological, on the etiological aspects of the disease. They also suggest the need of further studies aiming at correlating EEG data with the prognostic matters and the outcomes of the rehabilitative interventions.

## Figures and Tables

**Figure 1 brainsci-13-00345-f001:**
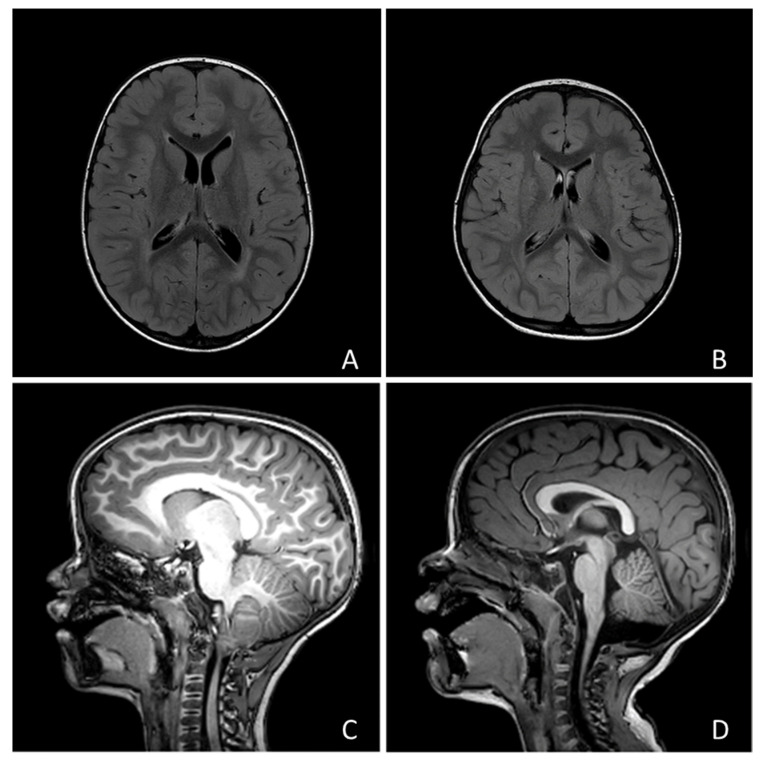
Four examples of the neuroradiological findings we reported in the text: (**A**) Macrocephaly and hyperintensity of peritrigonal white matter; (**B**) Mild asymmetry of lateral ventricles; (**C**) Chiari type I malformation; (**D**) Slight enlargement of Cisterna Magna.

**Figure 2 brainsci-13-00345-f002:**
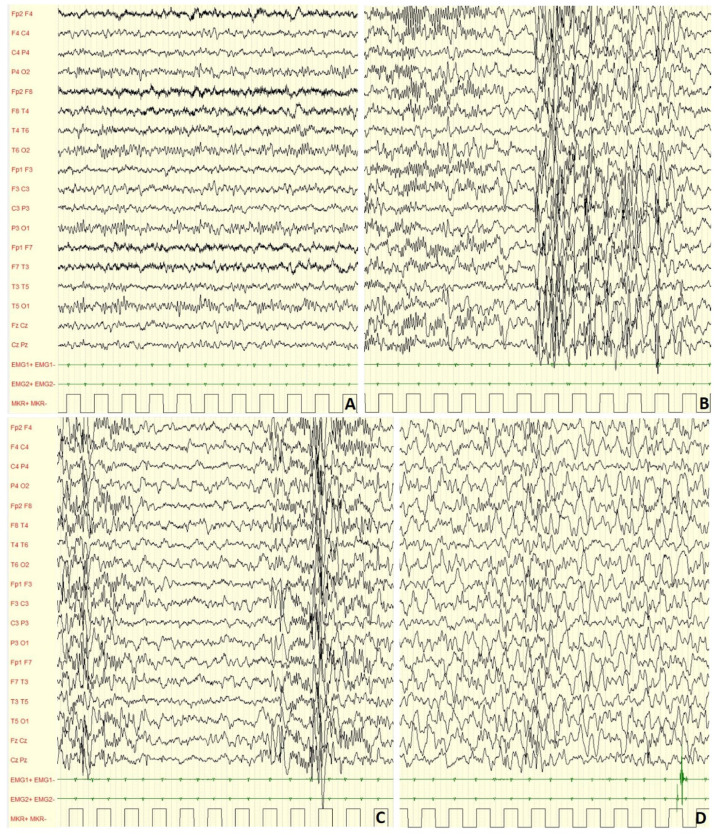
Male, 4 years-6 months PSG-EEG: (**A**) normal background activity in wakefulness; (**B**,**C**) PS-IEDs during NREM-2 phase of sleep; (**D**) reduction of PS-IEDs during NREM-3 phase of sleep.

**Figure 3 brainsci-13-00345-f003:**
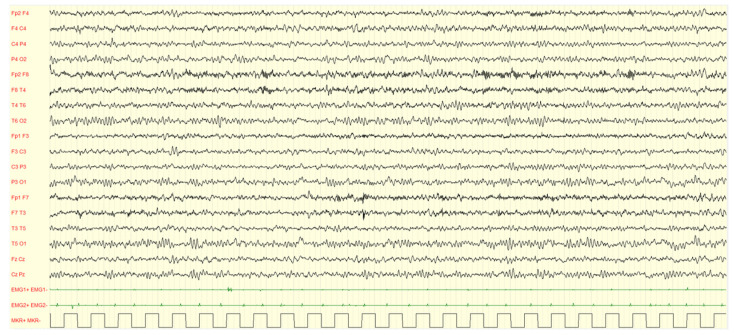
Male, 4 years PSG-EEG: background activity in wakefulness characterized by abundant rapid rhythms on the anterior regions.

**Figure 4 brainsci-13-00345-f004:**
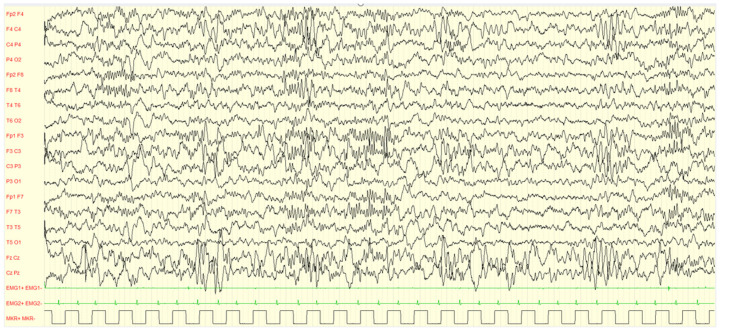
Male, 4 years PSG-EEG: background activity in sleep characterized by asynchronous and/or asymmetrical spindles and by K-complexes of irregular morphology with mixed paroxysmal slow waves.

**Figure 5 brainsci-13-00345-f005:**
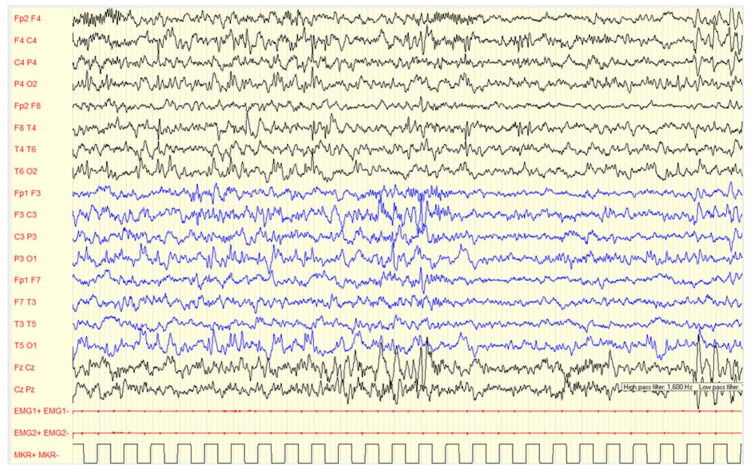
Female, 2 years-11 months, PSG (2–3 non-REM sleep): asynchronous sharp waves on right central-temporal and left frontal regions are showed. The background activity is characterized by excessive fast activity mainly over anterior areas, asynchronous sleep spindles.

**Table 1 brainsci-13-00345-t001:** Cohort data and results.

TOTAL SUBJECTS	292
M	248 (84.9%)
F	44 (15.1%)
MEAN AGE AT RECORDING (MONTHS)	34.6
DEVELOPMENTAL DELAY	190 (69.5%)
FEMALES with DEVELOPMENTAL DELAY	37 (84.0%)
MALES with DEVELOPMENTAL DELAY	153 (61.0%)

## Data Availability

The data presented in this study are available on request from the Psychopathology Department and the Clinical Neurophysiology Unit IRCCS E. Medea. The data are not publicly available in order to protect the children’s and their families’ privacy.

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
