# Peer review of "EEG Features in Autism Spectrum Disorder: A Retrospective Analysis in a Cohort of Preschool Children"

_brainsci, 2023, doi:10.3390/brainsci13020345_

Round 1
Reviewer 1 Report
Electroencephalography (EEG) and its application in autism spectrum disorder can further our understanding of the neurophysiology of this diagnosis and boost the overall knowledge of ASD. The topic is trendy, and some studies are focusing on this subject. Hence, I suggest the preferred terminology used consistently throughout the manuscript. Internationally acceptable wording indicates that people with permanent conditions instead of mentioning “Patient” are to be addressed as people with an autism spectrum disorder or intellectual disabilities, persons with an autism spectrum disorder or intellectual disabilities, children with autism spectrum disorder intellectual disabilities or adults with autism spectrum disorder intellectual disabilities.
[Line 87] In the clinical and instrumental investigations of ASD subjects, please make the following points clarify:
1. Were the children already diagnosed with ASD and referred to the study base for EEG? Or is the entire process of screening and diagnosis done in one place (if the answer is yes, what was the reason for re-evaluation?
2. Is this a secondary data analysis (EEG and diagnostic information are already collected in a clinic), or is it done for the purpose of the present study?
3. Are DSM5 criteria served as a type of screening, and ADOS2 and ADI-R served as the diagnostic tool? This is a more sensible approach)
4. Please explain the phrase “when needed” in line 91. Logically, ADOS 2 (based on the child behavior observation) covers the shortcoming of ADI-R, administered through the interview with the caregiver. Why did you use a different approach?
5. Coding and analyzing both ADOS2 and ADI-R requires special training and certification called reliability. Please provide more information about the professional who did the test administration.
6. I am unsure if the tests were applied in non-English communities (possibly Italy). If so, which version of the scales (for the ASD diagnosis and cognitive testing WPSI-III) was used? The validity and reliability of the scales in the community used are vital to support the findings.
It was helpful to know about the exclusion criteria, but how about the inclusion criteria? How many children or parents approached and wanted to be excluded? How did they inform about the study? This information will be beneficial.
In the results section, I did not understand what “exam” meant in sentences 124 and 125. Please provide more explanation.
I also find it very useful to understand why those subjects were excluded from the sample. Were they put aside because of the exclusion criteria?
Internationally the gender ratio for ASD is 1 to 4. Do you have any justification for your sample's 1 to 6 rate? I understand that the sampling approach was convenient, but even in a convenient sampling, the expectation is that the gender ratio is stable. Is there any data regarding the Italian ASD gender difference rate? (Because some countries reported different rates, such as 1 to 3).
The study has no limitation section, similar to any other study. There were uncontrollable factors that might have impacted the results, and they deserve to be mentioned at the end of the discussion part as some suggestions for further studies. As a limitation of the study, there are different factors regarding the children that have not been considered. For instance, the level of ASD functionality, which is possible to be determined by ADOS2, has been neglected, and ASD has been considered a unit diagnosis which is not. I also think that topics such as the impacts of gender, psychoeducational and socio-economical background of children and caregivers might have factors that need to be investigated more. The applicability of EEG in ASD diagnosis based on the presented findings and other studies might be another suggested topic for investigation and is correctly mentioned at the end of the conclusion section.
I also found the following minor point for correction:
page 1, line 43,
“In Italy, 1 in 43 87 children aged 7–9 years in Italy is affected by this disorder [4].”
Reviewer 2 Report
This study has important data that could help with understanding of the relationship between EEG abnormalities and ASD phenotype. However, the authors only provide high level prevalence data on these abnormalities that have already been described in the literature. The manuscript would significantly benefit from a more detailed analysis.
-There is no mention as to why this cohort of children underwent PSG/EEG. This may influence the results and should be discussed.
-Did the authors look in more detail about the type of delay (verbal vs nonverbal, if specific domains were more delayed than other domains)? It would be helpful to determine if the differences are in overall delays or if there are specific domains that are impacted.
-Any differences with the ADOS or level of severity of ASD?
-Need a reference for lines 206 and 207 stating that abnormal background activity is more associated with developmental delay.
-Since PSG was recorded, is there any particular phase of sleep that the abnormalities occurred? The authors have PSG data on all participants but did not provide any detail about the phase of sleep.
-Did the study intentionally exclude patients with epilepsy? That would explain the low percentage.
-It is not necessary to have the same information in the tables as in the text (e.g., Table 1).
-The y axis on the tables should be labelled.
Reviewer 3 Report
First, I would like to congratulate the authors on their interesting research idea. However, there are some points that should be further addressed:
1) The title should include the study design type and the main idea of the study.
2) In the abstract, a summary of the methodology should be provided. More details such as the study design type and the study setting should also be mentioned (private or public institution?). It should also be better explained the research question and how the present study relate to previously published literature in the abstract.
3) The authors should specify if the study setting is public or private, is it a Hospital or an outpatient clinic?
4) In the methods section, there are a few points that should be further clarified:
a) How was this study protocol chosen? Was it based on previous literature? What were the inclusion criteria? Was there a limit of age established for selection?
b) What was the IRB number of the study?
c) How was the power of the study calculated?
d) Was the data distributed normally?
e) Which correlation statistics were used? Graphs and statistical parameters should be provided.
f) A more detailed explanation of the statistics should be provided. For example, were t-tests performed? The use of statistics such as relative risk seems inadequate in this case since it is more suited for cohorts which does not seem to be the study design the authors conducted. I advise the authors to work with a professional statistician to improve their statistical analysis.
g) Is there an explanation for the sex prevalence on the study?
h) A table with baseline characteristics should be provided.
5) It would be interesting if the authors could provide a table with more details about their results and the clinical characteristics of their population ( age at diagnosis, physical exam abnormalities, years of following-up, medications used by the patients, comorbidities, psychiatric disorders, scholarity, prenatal or perinatal complications, prematurity, birth weight, more details on the diagnosis of ASD and developmental delay such as genetic assessments, metabolic and laboratory panels, neuroimaging apart from MRI, speech and language assessment, cognitive function assessments, multi-professional management strategies used, etc). Also, the authors should explain more about how seizures were classified, which classification and criteria were used, as well as provide a table with the clinical characteristics of the seizures they found (etiology, seizure type, epilepsy type). More clinical background information should also be provided especially in the cases where MRI was performed. Figures and graphs should be improved. The authors should also refrain from using pie charts as they can be misleading to readers. I advise searching for help from a professional statistician in order to improve the graphs and tables in this study.
6) The results of the genetic tests and laboratory workup should be provided.
7) It would be interesting if authors could provide a copy of EEGs on supplementary material. Also, neuroimaging of selected patients should also be provided, for example, the ones described in table 1.
8) The authors should try to use standardized abbreviations to improve the searchability of the article. For example, the abbreviation “PEA” should me removed and substituted by a standardized abbreviation.
9) The authors should improve their statistical analysis in order to be able to arrive at conclusions in their study.
10) More limitations of the study should be added to the discussion, such as those intrinsic to the retrospective design. Also, the authors should further explain why was there a low percentage of MRI, which other types of neuroimaging were performed instead, for example, CT scan.
11) There is a remarkably low percentage of epilepsy in this sample. The authors should provide further explanation about this fact and add the fact that there was also a low percentage of neuroimaging. Was videoEEG performed? Were at least 2 board-certified neurologists involved in the diagnosis of these patients?
12) The discussion should be improved and should include a dialogue between the present study and the previously published literature. What does this study add new?
13) Both the conclusions and discussion mention prognostication however there were not enough data about prognostication in the study. I suggest removing this sentence or including more details about prognostication in the introduction and statistical analysis.
14) The authors should work with English editing services to improve readability of the article.
Round 2
Reviewer 1 Report
This is an updated version of the previously submitted manuscript. Hence there are still some inconsistencies between the applied words such as patients, individuals with ASD, children, and subjects. (as an instance, the patient is still used in the text line 184 when the authors said: We excluded six patients from the study because they met the exclusion criteria.
The manuscript would benefit from a thorough copy edit, which will undoubtedly happen from the editorials, but the organization and writing are much improved.
Therefore, I am happy to recommend publication.
Reviewer 2 Report
The authors have done a good job addressing all of the feedback from the reviewer. The paper appears sufficient for publication.
Reviewer 3 Report
The authors have replied to the reviewer's comments sufficiently.
